# Efficient Equivariant High-Order Crystal Tensor Prediction via Cartesian Local-Environment Many-Body Coupling

**Dian Jin** [1 2]   **Yancheng Yuan** [3]   **Xiaoming Tao** [1 2]

## Abstract

End-to-end prediction of high-order crystal tensor properties from atomic structures remains challenging: while spherical-harmonic equivariant models are expressive, their Clebsch-Gordan tensor products incur substantial compute and memory costs for higher-order targets. We propose the Cartesian Environment Interaction Tensor Network (CEITNet), an approach that constructs a multi-channel Cartesian local environment tensor for each atom and performs flexible many-body mixing via learnable channel-space interactions. By performing learning in channel space and using Cartesian tensor bases to assemble equivariant outputs, CEITNet enables efficient construction of high-order tensor. Across benchmark datasets for order-2 dielectric, order-3 piezoelectric, and order-4 elastic tensor prediction, CEITNet surpasses prior high-order prediction methods on key accuracy criteria while offering high computational efficiency.

## 1. Introduction

Tensor properties of crystal materials are fundamental in advancing many technological fields, leading to significant innovations in device development. As a result, accurate prediction of crystal tensor properties has attracted increasing attention from the research community. First principles methods, such as density functional theory (DFT), have facilitated the prediction of various properties within an acceptable error margin compared to traditional laboratory experiments. However, density functional theory calculations can be computationally demanding, especially when evaluating high-order tensor properties of large crystalline systems, as they require iterative self-consistent optimization of both electronic states and atomic configurations with explicit wavefunction descriptions (Yan et al., 2024b). Their computational complexity scales cubically or even more steeply with the number of atoms in the system (Wu et al., 2005; Pathrudkar et al., 2024). Consequently, this leads to difficulties in high-order tensor prediction, limiting the development of related fields. On a broader level, the energy consumption and carbon footprint associated with large-scale computational chemistry and material simulations are garnering increasing attention (Schilter et al., 2024). Therefore, developing data-driven prediction methods that balance accuracy with efficiency is imperative for accelerating material screening and discovery.

A central challenge in applying deep learning to tensor prediction lies in satisfying geometric equivariance. Unlike predicting scalar properties like total energy or formation energy (Lupo Pasini et al., 2021; Davariashtiyani & Kadkhodaei, 2023), tensor properties exhibit strict directional dependency: when a crystal structure experiences rotation in 3D space, tensor properties must transform according to the coordinate transformation rules (Grisafi et al., 2018). Neglecting this inductive bias leads to physically inconsistent predictions and degraded sample efficiency and generalization capabilities (Wen et al., 2024), hindering the model's ability to learn correct physical laws from limited, expensive DFT data.

Currently, the dominant approach for achieving this geometric equivariance in deep learning community involves projecting geometric information onto irreducible representations of the rotation group using spherical harmonics, and then using Clebsch–Gordan (CG) tensor products to combine those irreps in an equivariant manner (Batatia et al., 2022; Villar et al., 2021; Geiger & Smidt, 2022; Heilman et al., 2024). While this paradigm shows great performance and physical consistency, it comes with a distinct cost: the tensor products and coupling of high-order representations are computationally demanding (Passaro & Zitnick, 2023). To mitigate the computational and memory burden, two alternative directions have emerged. The first approach utilizes canonicalization (Kaba et al., 2023; Hua et al., 2026),

[1]Research Institute for Intelligent Wearable Systems, The Hong Kong Polytechnic University [2]School of Fashion and Textiles, The Hong Kong Polytechnic University [3]Department of Applied Mathematics, The Hong Kong Polytechnic University. Correspondence to: Yancheng Yuan <yancheng.yuan@polyu.edu.hk>.

*Proceedings of the $43^{rd}$ International Conference on Machine Learning*, Seoul, South Korea. PMLR 306, 2026. Copyright 2026 by the author(s).

which maps the structure to a defined and unique canonical frame before employing a non-equivariant network for prediction, subsequently rotating the result back to the original coordinate system. While offering potential engineering speedups, canonicalization mappings may introduce discontinuities and numerical instabilities in degenerate or near-degenerate symmetry scenarios (Dym et al., 2024). Although continuous canonicalizations can be constructed in certain restricted scenarios, they are highly dependent on the alignment, thereby potentially compromising training stability and generalization when misaligned. The second approach involves Cartesian tensors, and replaces spherical tensor products with Cartesian tensor contractions and multiplications, and representative works have demonstrated the feasibility of this paradigm in molecular and tensor prediction tasks (Simeon & De Fabritiis, 2023; Wang et al., 2024a). However, existing methods often rely on stacking deeper tensor message-passing layers or explicitly enumerating angular information (Zhong et al., 2022; Choudhary & DeCost, 2021), leads to high computational cost and limited scalability. Furthermore, such approaches are seldom extended to end-to-end higher-order tensor prediction, and are mainly built for accurate machine-learning interatomic potentials (MLIPs).

Motivated by these limitations, we propose the Cartesian Environment Interaction Tensor Network (CEITNet). CEITNet constructs a multi-channel local environment by aggregating neighbor information on weighted Cartesian bases, where each channel encodes a distinct directional mode of a high-order geometric basis. It then employs a learnable channel-interaction matrix that couples environment channels, enabling flexible many-body mixing while maintaining computational efficiency. The atomic semantics and the associated coefficients are produced by an invariant message-passing network. This decoupled design improves efficiency by avoiding the propagation of high-order features. Experiments on multiple high-order tensor prediction benchmarks (order-2 dielectric tensor, order-3 piezoelectric tensor, and order-4 elastic tensor) demonstrate that CEITNet achieves state-of-the-art accuracy on key accuracy criteria while remaining computationally efficient.

**Conflict of Interest Disclosure**   The authors declare no financial or other substantive conflicts of interest that could reasonably be perceived to influence the work reported in this paper.

## 2. Preliminaries

### 2.1. Crystal Graph Construction

**Crystal structure.**   A crystal structure is defined by a unit cell containing a set of atoms, which repeats infinitely in three-dimensional space along three periodic lattice vectors.

It can be mathematically represented by the lattice matrix $\mathbf{L} = [\boldsymbol{l}_1, \boldsymbol{l}_2, \boldsymbol{l}_3] \in \mathbb{R}^{3 \times 3}$, which can be constructed from the lattice parameters $(a, b, c, \alpha, \beta, \gamma)$, together with the atomic sites $\{(Z_i, \mathbf{x}_i)\}_{i=1}^n$, where $Z_i$ denotes the atomic number and $\mathbf{x}_i = \mathbf{L}[x_i, y_i, z_i]^\top \in \mathbb{R}^3$ gives the Cartesian coordinates of the $i$-th atom within the unit cell (Wang et al., 2026; Hua & Lin, 2025).

**Periodic graph.**   We construct a periodic crystal graph $G$ with a cutoff radius $r_{\text{cut}}$ from lattice parameters and $n$ atomic sites in the unit cell. The node set $V$ corresponds to the atoms in the unit cell, where each node carries invariant atomic attributes $(Z_i)$. For each ordered pair $(j, i)$, if there exists a periodic image of atom $j$ whose distance to atom $i$ is within $r_{\text{cut}}$, we add a directed edge $(j \to i) \in E$, i.e., messages are passed from atom $j$ to atom $i$ (Yan et al., 2022). For each edge $(j \to i)$, we associate the displacement vector $\mathbf{r}_{ij} \in \mathbb{R}^3$, its length $d_{ij} = \|\mathbf{r}_{ij}\|$, and the direction encoding $\mathbf{n}_{ij} = \mathbf{r}_{ij}/\|\mathbf{r}_{ij}\|$, which will be used to construct Cartesian geometric bases in the tensor head. Note that $d_{ij}$ is invariant, whereas $\mathbf{r}_{ij}$ (and $\mathbf{n}_{ij}$) is equivariant.

### 2.2. High-Order Tensor Prediction

**Problem statement.**   Input a periodic crystal graph $G = (V, E)$, the goal is to learn a neural network model $f_\theta$ that maps the input graph to a Cartesian high-order tensor:

$$f_\theta: \ G \mapsto \mathbf{T},$$

where $\mathbf{T} \in \mathbb{R}^{3 \times \cdots \times 3}$ is an order-$r$ tensor representing material properties. Following Yan et al. (2024b) and (Hua et al., 2026), we focus on three specific tensor properties with orders $r \in \{2, 3, 4\}$, corresponding to the dielectric, piezoelectric, and elastic tensors, respectively.

Given a training set $\{(G^{(s)}, \mathbf{T}^{(s)})\}_{s=1}^S$, $S$ for number of samples, we learn the optimal parameters $\theta$ by minimizing a regression objective over the dataset:

$$\min_\theta \sum_{s=1}^S \mathcal{L}\Big(f_\theta(G^{(s)}), \mathbf{T}^{(s)}\Big),$$

where $\mathcal{L}(\cdot, \cdot)$ denotes regression loss that quantifies the prediction error.

**Equivariance.**   Beyond predictive accuracy, we require the model to satisfy geometric consistency. Since physical properties depend on the coordinate system, the predicted tensor must transform consistently with any rotation of the crystal structure. For any rotation/reflection $R \in \mathrm{O}(3)$, we require $\mathrm{O}(3)$ equivariance of order $r$:

$$f_\theta(R \cdot G) \ = \ R^{\otimes r} f_\theta(G),$$

where $R \cdot G$ denotes the rotated input graph, and $R^{\otimes r}$ represents the standard $r$-fold tensor product action of $R$ on a

order-$r$ Cartesian tensor. This constraint ensures that the physical laws described by the model remain consistent to the choice of reference frame.

## 3. Methods

The main architecture of CEITNet is shown in Fig. 1. To optimize computational efficiency, atomic information is captured by an invariant network. This invariant network is interchangeable with other invariant architectures, making the framework highly scalable. CEITNet also utilizes a flexible environment interaction matrix to form multi-body interactions, significantly enhancing expressivity. Specifically, CEITNet constructs high-order tensors via four steps: (1) learning invariant node representations, (2) constructing local geometric bases, (3) modeling many-body effects via local environment interactions, and (4) pooling to the final high-order tensor. The equivariance of CEITNet is demonstrated in Appendix A.7. The code is available at https://github.com/CLaSLoVe/ceitnet.

### 3.1. Invariant Encoding

The initial invariant representation of node $i$ is obtained via an embedding $\mathrm{Emb}_z$ of its atomic number $Z_i$:

$$\mathbf{h}_i^{(0)} = \mathrm{Emb}_z(Z_i).\tag{1}$$

Also, the geometric information of the edge is encoded as an invariant scalar feature. We employ a Radial Basis Function (RBF) expansion of the distance, followed by a Multi-Layer Perceptron (MLP) $\mathrm{MLP}_e$:

$$\mathbf{e}_{ij} = \mathrm{MLP}_e\Big(\mathrm{RBF}\big(g(d_{ij})\big)\Big),\tag{2}$$

where $g(\cdot)$ denotes a distance transformation function, and RBF projects the scalar distance onto a set of fixed basis functions.

To capture the $L$-hop chemical environment, we employ $L$ message passing layers (specifically, ComformerConv layers $\Phi_e$ (Yan et al., 2024a)):

$$\mathbf{h}_i^{(\ell)} = \Phi_e^{(\ell)}\left(\mathbf{h}_i^{(\ell-1)}, \left\{(\mathbf{h}_j^{(\ell-1)}, \mathbf{e}_{ij})\right\}_{j\in\mathcal{N}(i)}\right),\tag{3}$$

where $\mathcal{N}(i)$ represents the neighbors of node $i$. The final output $\mathbf{h}_i^{(L)}$ serves as an invariant representation of the local atomic information. See Appendix A.1 for more information about the invariant backbone.

### 3.2. Channelized Local Environment

To lift the scalar features into equivariant tensor representations, we construct local environment tensors from edge vectors.

**Bases construction.** For each directed edge $(j \to i)$ with relative displacement $\mathbf{r}_{ij}$, we first construct the geometric bases: the normalized direction vector $\mathbf{n}_{ij}$, the identity matrix $\mathbf{I}$, and the traceless deviatoric tensor $\mathbf{Q}_{ij} = \mathbf{n}_{ij}\mathbf{n}_{ij}^\top - \frac{1}{3}\mathbf{I}$. These bases serve as building blocks for our equivariant representations. Other bases can be constructed from these bases (e.g., $\mathbf{n}_{ij} \otimes \mathbf{I}$, $\mathbf{Q}_{ij} \otimes \mathbf{Q}_{ij}$, or $\mathbf{n}_{ij} \otimes \mathbf{Q}_{ij}$). All these bases form $\mathcal{B}_{ij}$. Crucially, since these bases are derived from geometric vectors, they inherently preserve rotational equivariance, ensuring that the learned representations respect the physical symmetry of the crystal system.

**Channelized local environment construction.** We combine the invariant representations with the equivariant bases to form a Channelized Local Environment. For each edge, we construct a context vector $\mathbf{z}_{ij} = [\mathbf{h}_i \,|\, \mathbf{h}_j \,|\, \mathbf{e}_{ij}]$ and generate $K$-dimensional channel weights via an MLP $\mathrm{MLP}_w$:

$$\mathbf{w}_{ij} = \mathrm{MLP}_w(\mathbf{z}_{ij}).\tag{4}$$

To ensure numerical stability across varying crystal densities, we introduce a learnable normalization $\deg(i)^{-p}$. The channelized local environment tensor $\mathbf{E}_i$ for node $i$ is defined as the weighted aggregation of geometric bases at this node $\mathbf{B}_{ij}$:

$$\mathbf{E}_{i,k} = \deg(i)^{-p} \sum_{j\in\mathcal{N}(i)} w_{ij,k} \cdot \mathbf{B}_{ij},\tag{5}$$

where $k = 1, \ldots, K$.

It is worth noting that we do not limit the node to a single environment tensor. Instead, we construct a set of environments by projecting edge features onto different geometric bases.

### 3.3. Equivariant Tensor Interaction Head

To synthesize the atomic output tensor from the channelized local environments, we introduce an equivariant tensor interaction head. The key design principle is to keep all learnable components in channel space, while generating tensors through equivariant bases. This yields a flexible equivariant mechanism to model high-order tensors while capturing many-body interactions. A general many-body coupling is given in Appendix A.5. We use three-body interaction here for computational efficiency and practical stability.

As $\mathbf{E}_i$ denote the compressed $K$-channel local environments for atom $i$, we define the atomic tensor generated from this step as $\mathbf{T}_i$:

$$\mathbf{T}_i = \mathcal{A}(\mathbf{E}_i^{\tau_0}) \,+\, \gamma \cdot \psi(\mathbf{E}_i^{\tau_L}, \mathbf{E}_i^{\tau_R}; \mathbf{M}) \,+\, \Delta\mathbf{T}_i.\tag{6}$$

where $\mathcal{A}(\cdot)$ is a linear projection path to gathering the direct impact of all channels from selected environment $\mathbf{E}_i^{\tau_0}$, $\psi(\cdot)$

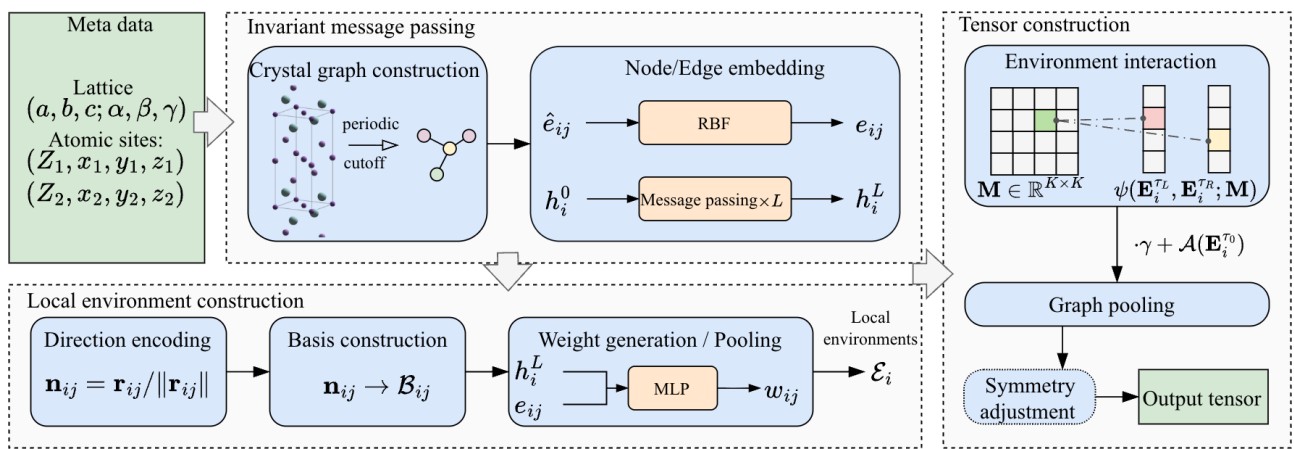

*Figure 1.* The architecture of CEITNet. It integrates an invariant message passing backbone with a Cartesian-based local environment constructor. The tensor construction head employs a learnable interaction matrix $\mathbf{M}$ to capture flexible many-body interactions. Note that some engineering details are omitted in the diagram for clarity.

is a coupling operator used for generating target tensors, and $\mathbf{M} \in \mathbb{R}^{K \times K}$ is a learnable channel interaction matrix. The scalar gate $\gamma$ controls the contribution of the coupling path for training stability. The residual term $\Delta \mathbf{T}_i$ is optional for task-specific components (e.g., global bias). The separable path and the coupling path may consume different environments, denoted by $\tau_0, \tau_L, \tau_R$.

Intuitively, $\mathcal{A}$ extracts a linear combination of tensor channels, while $\psi$ couples two environment streams by selecting the $m$-th channel of $\mathbf{E}_i^{\tau_L}$ and the $n$-th channel of $\mathbf{E}_i^{\tau_R}$, and conduct $\odot$, then weighted with $\mathbf{M}_{mn}$, as shown in the Fig. 1. Specifically, products of pooled neighbor expansions naturally induce higher-body interactions without explicit tuple enumeration, as the principle used in ACE/MACE (Batatia et al., 2022; Shapeev, 2016). Let $\odot$ denote a coupling, for simplicity, consider only channel $m$ and $n$ from two environments, and omit normalization term. Then

$$
\begin{aligned}
\mathbf{E}_{i,m}^{\tau_L} \odot \mathbf{E}_{i,n}^{\tau_R} &= \Big( \sum_{j \in \mathcal{N}(i)} w_{ij,m}^{\tau_L} \mathbf{B}_{ij}^{\tau_L} \Big) \odot \Big( \sum_{k \in \mathcal{N}(i)} w_{ik,n}^{\tau_R} \mathbf{B}_{ik}^{\tau_R} \Big) \\
&= \sum_{j \in \mathcal{N}(i)} \sum_{k \in \mathcal{N}(i)} w_{ij,m}^{\tau_L} w_{ik,n}^{\tau_R} \big( \mathbf{B}_{ij}^{\tau_L} \odot \mathbf{B}_{ik}^{\tau_R} \big).
\end{aligned}
$$
$$(7)$$

The cross-neighbor terms $(j, i, k)$ correspond to effective three-body (angular) correlations, without explicitly constructing triplets as input.

For dielectric prediction, we instantiate the coupling item:

$$
\psi(\mathbf{E}_i, \mathbf{E}_i; \mathbf{M}) = \sum_{m=1}^{K} \sum_{n=1}^{K} \mathbf{M}_{mn} \big( \mathbf{E}_{i,m} \cdot \mathbf{E}_{i,n} \big), \quad (8)
$$

where $\cdot$ here denotes standard matrix multiplication. The separable path is taken as $\mathcal{A}(\mathbf{E}_i) = \sum_k a_k \mathbf{E}_{i,k}$ with learnable coefficients $\{a_k\}$. Also, for dielectric tensors, an

isotropic background is added through $\Delta \mathbf{T}_i$ by adding $c\,\mathbf{I}$ with $c$ predicted from the global graph embedding. Finally, we enforce the intrinsic symmetry: $\epsilon_{ij} = \epsilon_{ji}$.

For elasticity, we instantiate the coupling term:

$$
\psi(\mathbf{E}_i, \mathbf{E}_i; \mathbf{M}) = \sum_{m=1}^{K} \sum_{n=1}^{K} \mathbf{M}_{mn} \big( \mathbf{E}_{i,m} \otimes \mathbf{E}_{i,n} \big), \quad (9)
$$

where $\otimes$ represents the tensor outer product, yielding a order-4 tensor. Distinct from the dielectric case, the separable path here is formulated as the sum of two terms, where the first term is an outer product of two independent linear combinations, and the second term is a linear combination of fourth-order environments: $\mathcal{A}(\mathbf{E}_i) = \Big( \sum_k u_k \mathbf{E}_{i,k}^{(2)} \Big) \otimes \Big( \sum_l v_l \mathbf{E}_{i,l}^{(2)} \Big) + \sum_{c \in \{qq, iq\}} w_c \mathbf{E}_{i,c}^{(4)}$, where $u_k, v_l, w_c$ are learnable weights, and $\mathbf{E}_{i,c}^{(4)} = \sum_{j \in \mathcal{N}(i)} w_{ij}^{(c)} \mathbf{B}_{ij}^{(c)}$, $\mathbf{B}_{ij}^{(qq)} = Q_{ij} \otimes Q_{ij}$, $\mathbf{B}_{ij}^{(iq)} = I \otimes Q_{ij} + Q_{ij} \otimes I$. The residual term $\Delta \mathbf{T}_i$ incorporates a bias parameterized by two Lamé coefficients $(c_1, c_2)$ predicted from the global crystal embedding. Finally, we enforce the intrinsic major and minor symmetries of the tensor (i.e., $C_{ijkl} = C_{jikl} = C_{ijlk} = C_{klij}$) on the output.

For piezoelectricity, the model needs interacting environments of different orders: vector channels $\mathbf{E}_i^{\text{vec}} \in \mathbb{R}^{K \times 3}$, matrix channels $\mathbf{E}_i^{\text{mat}} \in \mathbb{R}^{K \times 3 \times 3}$, and order-3 channels $\mathbf{E}_i \in \mathbb{R}^{K \times 3 \times 3 \times 3}$. The separable path simply aggregates the order-3 features via $\mathcal{A}(\mathbf{E}_i) = \sum_k a_k \mathbf{E}_{i,k}$. The coupling mechanism captures the interaction between $\mathbf{E}_i^{\text{vec}}$ and $\mathbf{E}_i^{\text{mat}}$ matrices:

$$
\psi(\mathbf{E}_i^{\text{vec}}, \mathbf{E}_i^{\text{mat}}; \mathbf{M}) = \sum_{m=1}^{K} \sum_{n=1}^{K} \mathbf{M}_{mn} \big( \mathbf{E}_{i,m}^{\text{vec}} \otimes \mathbf{E}_{i,n}^{\text{mat}} \big).
$$
$$(10)$$

Finally, the resulting tensor is symmetrized with respect to the last two indices to satisfy the piezoelectric tensor symmetry $e_{ijk} = e_{ikj}$.

Table 1 summarizes the specific instantiations of the interaction head for different material properties. When transferring to a new tensor property, the tensor head needs to be instantiated according to the target tensor order and intrinsic index symmetries. This design is localized to the output head, while the invariant backbone and channelized local-environment constructor remain unchanged.

*Table 1.* Example configuration of the interaction head.

| Task | Order | Coupling item |
|------|-------|---------------|
| Dielectric | 2 | $\mathbf{E} \cdot \mathbf{E}$ |
| Elasticity | 4 | $\mathbf{E} \otimes \mathbf{E}$ |
| Piezoelectric | 3 | $\mathbf{E}^{\text{vec}} \otimes \mathbf{E}^{\text{mat}}$ |

### 3.4. Readout

**Graph pooling.** To transition from atomic contributions to crystal properties, we employ a weighted pooling mechanism. The scalar importance score for each atom is computed as:

$$s_i = \mathbf{W}_2 \tanh(\mathbf{W}_1 \mathbf{h}_i), \qquad \alpha_i = \text{softmax}_{i \in \mathcal{V}_g}(s_i). \tag{11}$$

where $\mathbf{W}_1$ and $\mathbf{W}_2$ are two learnable matrices. The final crystal tensor is the weighted sum of atomic tensors:

$$\mathbf{T}_{\text{crystal}} = \sum_{i \in \mathcal{V}_g} \alpha_i \mathbf{T}_i. \tag{12}$$

**Symmetry adjustment.** After pooling, we optionally apply a symmetry-adjustment module. This is distinct from the intrinsic tensor symmetries, which are enforced by construction in our output parameterization. The motivation is, for an ideal crystal, the space-group symmetries impose additional component-wise constraints on the material tensor when expressed in a standard Cartesian coordinate system. While the network can learn these constraints implicitly to some extent, small residual violations may remain. Therefore, when the input corresponds to a perfect crystal, this post-processing step can improve physical consistency. We use the method proposed in Yan et al. (2024b) to construct a zero mask and an equality mask, which are applied to implement the symmetry adjustment. The zero and equality masks are inferred from the input crystal symmetry rather than from ground-truth tensor labels. Unless otherwise specified, the main benchmark tables report predictions after this symmetry-adjustment protocol.

## 4. Related Works

### 4.1. Equivariant Graph Neural Networks

Graph Neural Networks (GNNs), especially those based on the Message Passing Neural Network (MPNN) framework (Gilmer et al., 2017), have demonstrated strong performance in processing graph-structured data. With the development of GNNs, MLIPs have gradually attracted increasing attention. However, in the domain, it is necessary to incorporate inductive biases that enforce equivariance properties in the model architecture.

Early efforts focused on incorporating scalar–vector transformation (Schütt et al., 2021; Satorras et al., 2021; Thölke & Fabritiis, 2022). These methods maintain both scalar and three-dimensional vector features at each node, updating them jointly during message passing. They are computationally efficient and suitable for tasks involving directionally dependent quantities. In addition, another class of approaches achieves equivariant MLIPs by leveraging spherical harmonics and CG tensor products. Tensor Field Net (Thomas et al., 2018) and Cormorant (Anderson et al., 2019) pioneered this strategy by decomposing features into spherical harmonic components and using CG tensor products to achieve equivariant message passing. While highly expressive, these pipelines can incur non-trivial computational overhead due to repeated tensor products, which even worse when the representation order increases.

In recent years, an emerging paradigm has replaced explicit spherical-harmonic expansions and CG couplings with Cartesian tensor representations, leading to improved computational efficiency. Moment Tensor Potentials construct neighborhood moment tensors directly in Cartesian coordinates (Shapeev, 2016), Cartesian Atomic Cluster Expansion reformulates ACE-style expansions entirely in Cartesian space (Cheng, 2024), TensorNet adopts Cartesian tensor embeddings and simplifies feature mixing through matrix multiplications (Simeon & De Fabritiis, 2023), and HotPP extends node embeddings and messages to Cartesian tensors during message passing (Wang et al., 2024a). Work Zaverkin et al. (2024) introduces irreducible Cartesian tensors into message passing neural networks to improve the efficiency of equivariant molecular modeling. Complementarily, Shao et al. (2025) provides a systematic tool for decomposing Cartesian tensors into high-order irreducible components. These methods achieve high-accuracy MLIP, demonstrating the feasibility of using Cartesian representations to model physical systems with high predictive fidelity. While these equivariant architectures have been highly successful, many materials-physics observables of interest are naturally expressed as higher-order tensors. Accurately predicting such quantities requires models that can represent and generate higher-order tensors. This motivates the study of high-order tensor prediction within equivariant GNN

frameworks. However, this task remains challenging.

## 4.2. High-Order Tensor Prediction

A non end-to-end strategy is to learn an intermediate target such as the Hamiltonian (Wang et al., 2024b; Zhong et al., 2024) and then obtain high-order responses via post-processing. However, accurately predicting the Hamiltonian itself typically requires substantial high-fidelity data, and the subsequent post-processing can still be computationally expensive.

For end-to-end high-order tensor prediction, many existing approaches are based on spherical harmonics and irreducible representations. High-order tensors are obtained either by taking derivatives of learned tensors (Yan et al., 2024b) or by explicitly composing irreducible representations to assemble tensor outputs (Dong et al., 2025). These methods often incur high computational burden. Thus, researchers start to focus on using Cartesian representations to construct high-order tensors. Representative examples include ETGNN (Zhong et al., 2022), which explicitly enumerates edge triplets to incorporate three-body information when constructing third-order tensors, and GeoCTP (Hua et al., 2026), which adopts a canonicalization strategy that reduces the learning problem to an invariant backbone while enforcing equivariance through a deterministic alignment step, thereby substantially improving efficiency.

## 5. Experiments

### 5.1. Experimental Setup

**Datasets.** In this work, we used the datasets from GMT-Net, which is sourced from the JARVIS-DFT database (Choudhary et al., 2020). It includes dielectric, piezoelectric, and elastic tensors. The statistics are presented in Table 2. This dataset has been constructed with a keen emphasis on ensuring congruence between the properties and structures, achieved by extracting both the tensor property values and corresponding crystal structures directly from the DFT calculation files (Yan et al., 2024b).

Following the approach of Hua et al. (2026), we removed the zero tensors in the piezoelectric dataset. This step is necessary because keeping these zeros can negatively affect model training. If we don't remove them, a simple model that always predicts zero could achieve near-SOTA results, which makes the evaluation unreliable, this is also reported in Hua et al. (2026). See Appendix A.2 for more details about this.

**Baseline methods.** We selected several state-of-the-art methods designed for high-order tensor prediction, including ETGNN (Zhong et al., 2023), GMTNet (Yan et al., 2024b), and GeoCTP (Hua et al., 2026), as our baselines.

*Table 2.* Dataset statistics for different order tensor prediction. Mean and STD are calculated from Frobenius norm.

| Dataset | Order | # Samples | Mean | STD | Unit |
|---|---|---|---|---|---|
| Dielectric | 2 | 4713 | 14.7 | 18.2 | – |
| Piezoelectric | 3 | 2701 | 0.79 | 4.03 | $C/m^2$ |
| Elastic | 4 | 14220 | 327 | 249 | GPa |

For the model architecture, we adopt the same backbone ComformerConv (Yan et al., 2024a) that used by both GMT-Net and GeoCTP for the invariant message-passing component to ensure a fair comparison. See Appendix A.2 for more information about the details and configurations of these models.

**Evaluation metrics.** For fair comparison, we follow the exact evaluation metrics adopted in Yan et al. (2024b) and Hua et al. (2026). Specifically, we report the mean Frobenius-norm distance (Fnorm), computed as the Frobenius norm of the difference between the predicted tensor and the ground-truth tensor, averaged over the test set. This metric provides an $O(3)$-invariant measure for crystal tensor properties, capturing the overall geometric discrepancy between predictions and references regardless of global rotations. We also report the high-quality prediction rate (EwT $m\%$), defined as the fraction of test samples satisfying:

$$\frac{\text{Fnorm(error)}}{\text{Fnorm(label)}} < m\%.$$

This threshold is consistent with DFPT standards for comparison against experimental measurements (Petousis et al., 2016; Yan et al., 2024b), and serves as a robust and physically meaningful metric for prediction accuracy. It measures the proportion of samples whose predictions are within an application-relevant relative tolerance. The sizes of each test dataset are detailed in Table.3.

*Table 3.* Test set sizes for each task under the GMTNet split.

| Task | Dielectric | Piezoelectric | Elastic |
|---|---|---|---|
| # Instances | 471 | 270 | 1422 |

**Experimental settings.** For fair comparison, we follow the exact experimental settings of Yan et al. (2024b) and Hua et al. (2026): For each property, we split the samples into training, evaluation, and test sets using ratio 8:1:1. To train our model, we use Huber loss with AdamW, $10^{-5}$ weight decay, and polynomial decay for the learning rate.

The experiments were carried out with the PyTorch framework (Paszke et al., 2019) on a server running Ubuntu 22.04.5 LTS. We utilized a single NVIDIA A100 GPU for training and evaluation. See Appendix A.2 for more detailed settings.

# 6. Experimental Results

## 6.1. Performances on High-Order Tensor Benchmarks

**Predicting dielectric tensors.**   The performance of various models in predicting the dielectric tensor is summarized in Table 4. The results of ETGNN and GMTNet are obtained from Yan et al. (2024b), those of GeoCTP from Hua et al. (2026).

As shown in Table 4, our method consistently achieves the best performance across all metrics. In terms of Fnorm, our method significantly reduced the error, indicating a more accurate reconstruction of the overall tensor structure. As for the EwT metrics, for all three thresholds, our method show significantly higher accurate rates than all existing baselines, demonstrating its robustness in delivering highly reliable and precise predictions.

*Table 4.* Comparison on the dielectric dataset. Lower Fnorm and higher EwT indicate better performance. Best results are highlighted in bold, and second-best results are underlined. Results are obtained from the originally reported numbers.

| Method | Fnorm ↓ | EwT 25% ↑ | EwT 10% ↑ | EwT 5% ↑ |
|--------|---------|-----------|-----------|----------|
| ETGNN  | 3.92    | 81.3%     | 41.6%     | 23.8%    |
| GMTNet | 3.50    | 84.5% | 57.1% | 27.8%    |
| GeoCTP | 3.23 | 83.2%     | 56.8%     | 35.5% |
| Ours   | **2.87** | **86.1%** | **63.8%** | **39.3%** |

**Predicting piezoelectric tensors.**   The performance of various models in predicting the piezoelectric tensor is summarized in Table 5. Results for ETGNN, GMTNet, and GeoCTP are from Hua et al. (2026).

Compared to dielectric tensors, predicting the piezoelectric tensor is substantially more challenging.   As shown in Table 5, our method achieves clear improvements across all evaluation metrics. As for the EwT metrics, for all three thresholds, our method show significantly higher accurate rates than all existing baselines. This indicates that our approach not only significantly reduces the absolute error, but also substantially reduces the relative error.

*Table 5.* Comparison on the Piezoelectric dataset. Lower Fnorm and higher EwT indicate better performance. Best results are highlighted in bold, and second-best results are underlined. Results are obtained from the originally reported numbers.

| Method | Fnorm ↓ | EwT 25% ↑ | EwT 10% ↑ | EwT 5% ↑ |
|--------|---------|-----------|-----------|----------|
| ETGNN  | 0.873   | 0.00%     | 0.00%     | 0.00%    |
| GMTNet | 0.752 | 6.29%  | 1.48% | 1.11% |
| GeoCTP | 0.778   | 2.59%     | 1.14%     | 0.04%    |
| Ours   | **0.517** | **21.98%** | **5.80%** | **2.72%** |

**Predicting elastic tensors.**   The performance of various models in predicting the elastic tensor is summarized in Table 6. The results of GMTNet are taken from Yan et al. (2024b). Notably, the original GeoCTP paper uses a different dataset and evaluation protocol. Therefore, we re-implemented GeoCTP, and results reported in this benchmark for both ETGNN and GeoCTP were produced by us. See Appendix A.2 for more information.

As shown in Table 6, in terms of Fnorm, our method provides competitive results comparable to state-of-the-art baselines. For the EwT metrics, our method consistently achieves the best overall performance. This indicates that our approach achieves competitive performance in terms of absolute error, while substantially reducing the relative error. We also analyze the per-sample error distribution using a CEITNet checkpoint with Fnorm = 69.4. The error distribution is long-tailed: the worst 5% of test samples contribute 32.8% of the total Fnorm. This indicates that the mean Fnorm is disproportionately affected by a small number of hard outliers, whereas the stronger EwT scores indicate better relative accuracy on a larger fraction of the test set. Thus, the remaining Fnorm gap is more consistent with a tail-risk issue than with uniform degradation across samples.

*Table 6.* Comparison on the Elastic dataset. Lower Fnorm and higher EwT indicate better performance. Best results are highlighted in bold, and second-best results are underlined. Results are taken from original papers when available on the same dataset/protocol; otherwise (due to missing such results) we re-evaluate under the same protocol ($^*$).

| Method | Fnorm ↓ | EwT 25% ↑ | EwT 10% ↑ | EwT 5% ↑ |
|--------|---------|-----------|-----------|----------|
| ETGNN$^*$  | 102.76  | 49.6%     | 9.8%      | 2.3%     |
| GMTNet | **67.38** | 66.1% | 21.8%  | 7.7%     |
| GeoCTP$^*$ | 85.00  | 59.6%     | 23.9% | 10.4% |
| Ours   | 70.11 | **70.6%** | **32.2%** | **14.4%** |

## 6.2. Efficiency

Beyond predictive accuracy, inference runtime efficiency is also important considerations in high-order material tensor modeling, especially for applications such as high-throughput screening. The results is shown in Table 7.

We further assess parameter efficiency on the dielectric task. Our model is more compact than prior baselines, using only 0.6M trainable parameters versus 1.1M for ETGNN and 0.7M for GMTNet, reductions of approximately 46% and 14%, respectively. Despite this smaller model size, CEITNet achieves lower Fnorm error, indicating improved parameter efficiency without sacrificing predictive accuracy. This efficiency improvement can be attributed to two key design choices: first, the decoupling of invariant encoding from equivariant tensor construction ensures that geometric in-

formation is introduced only when necessary, avoiding the propagation of high-order equivariant features throughout the entire network; second, the tensor interaction head effectively controls computational complexity while keeping expressive.

*Table 7.* Comparison of runtime/throughput performance during inference periods (both w/o symmetry correction module, runtime is rounded to the nearest second).

|  | Dielectric | Piezoelectric | Elastic |
|---|---|---|---|
| # item | 471 | 270 | 1422 |
| GMTNet Time (s) ↓ | 21 | 50 | 616 |
| GeoCTP Time (s) ↓ | 6 | 5 | 50 |
| CEITNet Time (s) ↓ | 5 | 6 | 46 |
| GMTNet item/s ↑ | 22.02 | 5.40 | 2.31 |
| GeoCTP item/s ↑ | 69.61 | 52.08 | 28.39 |
| CEITNet item/s ↑ | 85.90 | 40.64 | 30.79 |

## 6.3. Ablation Study

To understand the role of each design choice in CEITNet, we run an ablation study on the dielectric benchmark (Table 8). The complete model performs best overall, achieving the lowest average error (Fnorm) while also maintaining strong high-precision results (EwT). This result suggests that all components are important in predicting high-order tensors.

We analyze channel interaction (w/o mix) by setting the channel-interaction matrix to the identity. This results in a clear performance degradation, indicating that allowing information exchange across channels is important. Next, removing the interaction term while keeping the separable path (w/o inter) noticeably hurts performance. This implies that the information carried by interaction term is different from that by separable path. It introduces important coupling that helps to represent environment interactions. After removing the separable path (w/o sep), the average error changes little, while all EwT metrics drop moderately. This suggests that the separable path mainly provides a coarse, direct signal from local environments. We also ablate the normalization (w/o norm). The result indicates that normalization contributes to the training performance.

We additionally study the effect of channel width $K$. Intermediate widths yield the most robust behavior, while very small $K$ limits representational capacity and increases average error. On the other hand, overly large $K$ does not translate into better accuracy and can even make optimization harder, and the mixing matrix becomes large and harder to train. These results suggest a practical trade-off between expressivity and training stability. In this setting, $K{=}16$ offers the best overall balance. See Appendix A.2 for more information.

*Table 8.* Ablation study on dielectric tensor prediction. Best results are highlighted in bold, and second-best results are underlined. Our CEITNet use $K = 16$.

| Setting | Fnorm | EwT 25% | EwT 10% | EwT 5% |
|---|---|---|---|---|
| CEITNet | **2.87** | 86.1% | 63.8% | 39.3% |
| w/o mix | 3.05 | 83.9% | 61.8% | 35.9% |
| w/o inter | 3.21 | 86.2% | 62.0% | 37.5% |
| w/o sep | 2.91 | 85.8% | 62.0% | 35.2% |
| w/o norm | 3.23 | 84.3% | 52.9% | 29.1% |
| $K{=}4$ | 3.16 | 87.6% | 63.9% | 37.5% |
| $K{=}8$ | 3.04 | 86.4% | 58.6% | 35.2% |
| $K{=}32$ | 3.22 | 83.7% | 60.7% | 35.9% |
| $K{=}64$ | 3.22 | 85.4% | 63.9% | 35.9% |

*Table 9.* Predicting symmetry-constrained zero-valued dielectric tensor elements. Success rate measured by error $< 10^{-5}$. Property labels in the curated dataset achieve 100% success rate.

| Crystal System | GMTNet | | CEITNet (Ours) | |
|---|---|---|---|---|
|  | w/o corr. | corr. | w/o corr. | corr. |
| Cubic | 44.6% | 100% | 73.0% | 100% |
| Tetragonal | 56.6% | 100% | 59.2% | 100% |
| Hexa-Trigonal | 57.0% | 100% | 59.1% | 100% |
| Orthorhombic | 68.7% | 100% | 71.1% | 100% |
| Monoclinic | 78.2% | 100% | 78.2% | 100% |

## 6.4. Symmetry Constraints

Following prior work (Yan et al., 2024b), we optionally apply a symmetry correction module at inference time. To quantify how well the models respect symmetry without explicit correction, we follow the protocol of GMTNet and evaluate the success rate on symmetry-constrained zero-valued dielectric tensor elements. A prediction is counted as successful if absolute error $< 10^{-5}$. Table 9 reports success rates across crystal systems. Without correction, CEITNet already satisfies a substantially larger fraction of symmetry-enforced zeros, indicating that the proposed Cartesian environment construction and channel interaction head learn a more physically consistent tensor structure. After applying the symmetry correction module, both CEITNet and GMTNet achieve 100% success.

## 7. Conclusion

In conclusion, we propose the Cartesian Environment Interaction Tensor Network, for predicting high-order crystal tensor properties. To improve computational efficiency, CEITNet adopts a decoupled design that separates invariant atomic encoding from equivariant tensor construction, avoiding the propagation of high-order equivariant features. Building on Cartesian geometric bases derived from interatomic directions, CEITNet constructs channelized local-environment tensors and introduces a learnable channel-

interaction module to enable flexible many-body mixing. Experimental results demonstrate that CEITNet achieves state-of-the-art performance across different high-order tensor prediction tasks on key accuracy criteria. Moreover, compared with spherical-harmonic-based approaches, CEITNet offers substantial computational advantages, highlighting its effectiveness and flexibility in multi-task settings.

Despite the strong performance of CEITNet, there are several limitations: First, compared with equivariant methods based on spherical harmonics, our Cartesian framework typically requires non-trivial design of the corresponding coupling process when transferring to different tensor types. Systematically selecting a set of bases and coupling templates that is both computationally efficient and sufficiently expressive for a new task still involves some hyperparameter tuning. Second, in this work, we instantiate the model with a coupling between two environment streams, which effectively restricts the explicit interaction order to three-body information, which is also efficient in most cases. In principle, the framework can be extended to model higher-order many-body effects, however, higher-order interactions inevitably introduce trade-offs in computational cost, numerical stability, and the risk of overfitting.

Future work may explore more systematic task transfer, controlled extensions to higher-order many-body couplings, and stronger symmetry-aware inductive biases without sacrificing efficiency.

## Acknowledgments

This research was supported by the Research Grants Council of Hong Kong (Grant No. T42-513/24-R). Yancheng Yuan was supported by the NSFC Young Scientists Fund (Grant No. 12501440) and RGC Early Career Scheme (Project No. 25305424).

## Impact Statement

This paper presents work whose goal is to enable efficient and accurate prediction of high-order physical property tensors in crystal materials. There are many potential societal consequences of our work, none of which we feel must be specifically highlighted here.

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

# A. Appendix

## A.1. Invariant Backbone

Our model uses exact the message passing part (ComformerConv) of the eComFormer (Yan et al., 2024a), which is same as the GMTNet and GeoCTP for fairness. The ComformerConv is primarily implemented through an attention mechanism and a message-passing mechanism, as illustrated in Fig. 2.

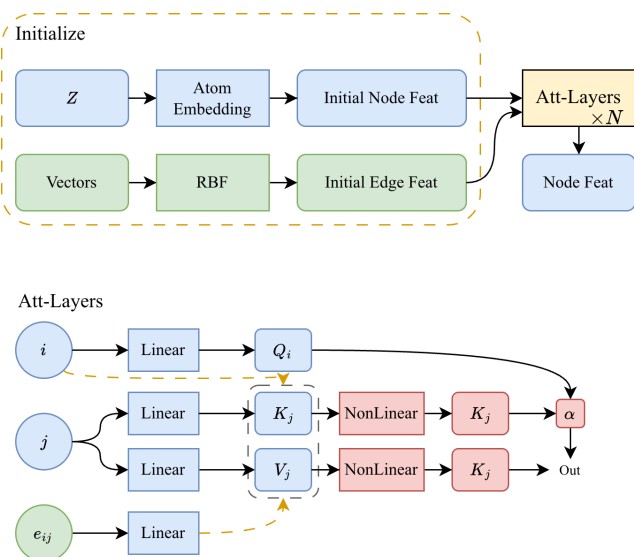

*Figure 2.* The architecture of ComformerConv.

## A.2. Reproducibility

**Datasets.** We use the curated JARVIS-DFT tensor datasets and the GMTNet split with train/val/test ratio 8:1:1 for each task. For piezoelectric tensors, we follow the GeoCTP preprocessing protocol and remove zero tensors, as nearly 50% samples of the dataset are zero, which would otherwise make trivial predictors overly competitive. As shown in Table 10, when zero tensors are not excluded, a naive Zero Func baseline (predicting an all-zero tensor) already achieves strong EwT scores, indicating that the metric can be dominated by symmetry-enforced zeros rather than the model's ability to recover non-trivial tensor components.

*Table 10.* Comparison on the Piezoelectric dataset if not exclude zero tensors.

| Method | Fnorm ↓ | EwT 25% ↑ | EwT 10% ↑ | EwT 5% ↑ |
|---|---|---|---|---|
| GMTNet | 0.37 | 49.1% | 46.3% | 45.7% |
| Zero Func | 0.43 | 45.7% | 45.7% | 45.7% |

**Training hyperparameters.** To ensure a fair comparison, all methods share identical crystal graph construction and training protocols. The training configurations are standardized across all tasks—including dielectric, piezoelectric, and elastic tensor predictions. Specifically, we train the models for 200 epochs using a batch size of 64. Optimization is performed using the AdamW optimizer (weight decay of $10^{-5}$) with an initial learning rate of 0.001, following a polynomial decay schedule. Huber loss is employed as the objective function, and the optimal checkpoint for each method is selected based on the same validation metric.

**Runtime benchmarking.** Runtime/throughput is measured on the test set for all methods under the same hardware and software environment, using the same batch size and graph construction parameters. To isolate the core model efficiency, we disable the optional symmetry-correction module for all methods during runtime evaluation.

**Configurations of CEITNet.** Crystal graphs are constructed using a radius-based cutoff determined by the 16th nearest neighbor. For edge embeddings, we employ Radial Basis Function (RBF) kernels with values ranging from -4 to 0, mapping the term $-c/\|v_{ij}\|_2$ (where $c = 0.75$) to a 512-dimensional vector. The model architecture consists of $L = 2$ message-passing layers with a hidden dimension of $H = 128$. The number of channels is set to $K = 16$ for all benchmarks. We keep the experimental protocol consistent with that used for the other baselines. The results are conducted over three runs, the results of mean $\pm$ standard deviation are shown in Table 11.

*Table 11.* CEITNet results of mean $\pm$ standard deviation over three runs.

| Method | Fnorm $\downarrow$ | EwT 25% $\uparrow$ | EwT 10% $\uparrow$ | EwT 5% $\uparrow$ |
|---|---|---|---|---|
| Dielectric | 2.87±0.20 | 86.1±2.0% | 63.8±0.3% | 39.3±1.8% |
| Piezoelectric | 0.517±0.01 | 22.0±3.6% | 5.80±0.8% | 2.72±0.6% |
| Elastic | 70.1±0.67 | 70.6±1.1% | 32.2±0.9% | 14.4±0.7% |

**Configurations of ETGNN and GMTNet.** For ETGNN and GMTNet, we follow the exact experimental protocol of Yan et al. (2024b). Specifically, we adopt the same curated datasets, data splits, preprocessing, and evaluation metrics, and we utilize the official codebases and the originally reported results from Yan et al. (2024b) for the corresponding benchmark settings.

**Configurations of GeoCTP.** For GeoCTP, we use the originally reported results when they are available under a directly comparable dataset/protocol setting (dielectric and piezoelectric tasks). For the elastic tensor task, the results reported in the GeoCTP paper are obtained under a different dataset, and are therefore not directly comparable to the GMTNet benchmark used in this work. To enable a fair comparison on the elastic task, we re-implement GeoCTP and evaluate it under the GMTNet split and the same preprocessing and metrics as other methods.

## A.3. Additional Ablations

We additionally evaluate some ablation studies on piezoelectric and elastic tensors datasets. The results are shown in Table. 12 and 13.

*Table 12.* Ablation on the piezoelectric benchmark.

| Piezoelectric | Fnorm $\downarrow$ | EwT 25% $\uparrow$ | EwT 10% $\uparrow$ | EwT 5% $\uparrow$ |
|---|---|---|---|---|
| K=8 | 0.536 | 21.48% | 4.07% | 1.11% |
| K=16 | 0.517 | **21.98%** | **5.80%** | **2.72%** |
| K=32 | **0.514** | 20.00% | 5.19% | 1.85% |
| w/o mix | 0.539 | 17.04% | 4.81% | 1.85% |
| w/o inter | 0.567 | 14.07% | 4.07% | 1.11% |
| w/o sep | 0.693 | 6.67% | 1.85% | 1.48% |

*Table 13.* Ablation on the elastic benchmark.

| Elastic | Fnorm $\downarrow$ | EwT 25% $\uparrow$ | EwT 10% $\uparrow$ | EwT 5% $\uparrow$ |
|---|---|---|---|---|
| K=8 | 71.23 | 69.90% | 31.50% | **14.56%** |
| K=16 | **70.11** | **70.60%** | **32.20%** | 14.40% |
| K=32 | 72.73 | 69.34% | 29.89% | 12.80% |
| w/o mix | 71.10 | 70.50% | 31.60% | 13.40% |
| w/o inter | 72.23 | 69.34% | 29.61% | 12.03% |
| w/o sep | 85.65 | 57.74% | 17.65% | 7.03% |

## A.4. Additional Generalization Experiments

**External Materials Project dielectric benchmark.** We evaluate CEITNet on the Materials Project dielectric benchmark using the same split protocol. Table 14 shows that CEITNet also outperforms GMTNet on this external benchmark.

*Table 14.* Results on the Materials Project dielectric benchmark.

| Method | Fnorm ↓ | EwT 25% ↑ | EwT 10% ↑ | EwT 5% ↑ |
|---|---|---|---|---|
| GMTNet | 0.202 | 100% | 89.3% | 58.1% |
| CEITNet | **0.188** | 100% | **91.4%** | **62.4%** |

*Table 15.* Leave-one-crystal-system-out results on the dielectric benchmark.

| Method | Held-out system | Fnorm↓ | EwT 25%↑ | EwT 10%↑ | EwT 5%↑ |
|---|---|---|---|---|---|
| GMTNet | Cubic | 11.1 | 61.5% | 31.5% | 16.1% |
| | Orthorhombic | 4.2 | 71.9% | 38.9% | 13.0% |
| | Hexagonal | 6.6 | 76.8% | 49.1% | 22.3% |
| | Trigonal | 5.8 | 77.7% | 33.0% | 13.2% |
| | Monoclinic | 3.4 | 79.3% | 43.4% | 14.7% |
| | Tetragonal | 4.3 | 82.4% | 42.8% | 17.6% |
| CEITNet | Cubic | 10.7 | 64.6% | 37.9% | 20.7% |
| | Orthorhombic | 3.6 | 78.2% | 41.8% | 17.2% |
| | Hexagonal | 5.7 | 80.7% | 56.9% | 29.8% |
| | Trigonal | 5.6 | 81.1% | 56.2% | 29.5% |
| | Monoclinic | 3.3 | 83.6% | 49.8% | 21.6% |
| | Tetragonal | 4.1 | 85.0% | 47.9% | 20.5% |

**Leave-one-crystal-system-out evaluation.** We further conduct leave-one-crystal-system-out experiments on the dielectric benchmark. For each split, one crystal system is held out for testing and the model is trained on the remaining systems. Table 15 shows that CEITNet consistently outperforms GMTNet across all six held-out systems.

## A.5. General Many-Body Coupling

CEITNet can be written as a general many-body coupling framework over pooled local-environment streams. Let $\mathbf{E}_i^\tau$ denote a $K$-channel local environment constructed from basis type $\tau$. A general atomic tensor head can be expressed as

$$\mathbf{T}_i = \mathcal{A}(\mathbf{E}_i^{\tau_0}) + \sum_{q=2}^{q_{\max}} \gamma_q \psi_q\Big(\mathbf{E}_i^{\tau_{q,1}}, \ldots, \mathbf{E}_i^{\tau_{q,q}}; \mathbf{M}^{(q)}\Big) + \Delta\mathbf{T}_i. \tag{13}$$

Here $\mathcal{A}(\cdot)$ is a separable projection path, $\psi_q(\cdot)$ is a channel-space coupling operator over $q$ environment streams, $\mathbf{M}^{(q)}$ denotes learnable channel-interaction parameters, and $\Delta\mathbf{T}_i$ contains optional task-specific residual components. Expanding each pooled environment into a neighbor summation gives products over $q$ neighbor indices, so a $q$-stream coupling induces effective $(q+1)$-body interactions centered at atom $i$. In this work, we use $q_{\max} = 2$, which induces effective three-body interactions $(j, i, k)$:

$$\mathbf{E}_{i,m}^{\tau_L} \odot \mathbf{E}_{i,n}^{\tau_R} = \sum_{j \in \mathcal{N}(i)} \sum_{k \in \mathcal{N}(i)} w_{ij,m}^{\tau_L} w_{ik,n}^{\tau_R} \left( \mathbf{B}_{ij}^{\tau_L} \odot \mathbf{B}_{ik}^{\tau_R} \right). \tag{14}$$

Higher-order couplings with $q > 2$ are possible in principle, but they increase the number of channel-interaction parameters and the cost of tensor assembly. We therefore restrict the current instantiation to $q = 2$ to balance expressivity, efficiency, and training stability.

## A.6. Local Environment Coupling

**Density trick.** Our many-body mixing follows the density-product principle used in ACE/MACE (Batatia et al., 2022): taking products of pooled (summed) neighbor expansions naturally induces higher-body interactions without explicit tuple enumeration. In contrast to MACE, which builds equivariance via spherical harmonics and explicit CG couplings, CEITNet adopts Cartesian bases together with a flexible interaction matrix, keeping learnable transformations confined to channel space.

Recall that for each atom $i$, CEITNet constructs a $K$-channel local environment tensor by aggregating over neighbors:

$$\mathbf{E}_{i,k} = \sum_{j \in \mathcal{N}(i)} w_{ij,k}\, \mathbf{B}_{ij}, \tag{15}$$

where $w_{ij,k}$ are scalar channel weights predicted from invariant features, and $\mathbf{B}_{ij}$ is an equivariant Cartesian basis tensor (e.g., built from $\mathbf{n}_{ij}$, $\mathbf{Q}_{ij}$, and $\mathbf{I}$).

Consider the coupling term used in our interaction head. For simplicity, denote a generic operator $\odot$ that produces a tensor of desired order (e.g., matrix multiplication or outer product), and define

$$\psi(\mathbf{E}_i, \mathbf{E}_i; \mathbf{M}) = \sum_{m=1}^{K} \sum_{n=1}^{K} \mathbf{M}_{mn}\, (\mathbf{E}_{i,m} \odot \mathbf{E}_{i,n}). \tag{16}$$

Expanding $\mathbf{E}_{i,m} \odot \mathbf{E}_{i,n}$ yields

$$\mathbf{E}_{i,m} \odot \mathbf{E}_{i,n} = \Big( \sum_{j \in \mathcal{N}(i)} w_{ij,m}\, \mathbf{B}_{ij} \Big) \odot \Big( \sum_{k \in \mathcal{N}(i)} w_{ik,n}\, \mathbf{B}_{ik} \Big) \tag{17}$$

$$= \sum_{j \in \mathcal{N}(i)} \sum_{k \in \mathcal{N}(i)} \big( w_{ij,m}\, w_{ik,n} \big)\, (\mathbf{B}_{ij} \odot \mathbf{B}_{ik}). \tag{18}$$

The double summation over $(j, k)$ indicates that the coupling term contains cross-neighbor interactions around the center atom $i$, corresponding to effective three-body contributions of the form $(j, i, k)$, without explicitly enumerating triplets in the network input. Importantly, the weights $w_{ij,m}$ and $w_{ik,n}$ are learnable functions of chemistry and distances, enabling data-driven modulation of these many-body effects.

### A.7. Equivariance Proof

We prove that the CEITNet is O(3)-equivariant. i.e., for an order-$r$ prediction ($r \in \{2, 3, 4\}$) and any orthogonal transform $R \in \mathrm{O}(3)$ applied to the input structure, the model output satisfies

$$\mathrm{CEITNet}(R \cdot G) \;=\; \rho_r(R)\, \mathrm{CEITNet}(G), \qquad \rho_r(R) := R^{\otimes r}. \tag{19}$$

Throughout, we follow the notations in the main text.

**Lemma A.1** (Equivariance of the basis). *For a basis tensor of order $q$, the basis used in the head satisfies*

$$\mathbf{B}(R\mathbf{n}) \;=\; \rho_q(R)\, \mathbf{B}(\mathbf{n}), \qquad \forall R \in \mathrm{O}(3). \tag{20}$$

*Proof.* Since $\mathbf{n} \mapsto R\mathbf{n}$, any basis element can be written as a linear combination of tensors of the form $\mathbf{B}(\mathbf{n}) = \mathcal{P}(\mathbf{n}^{\otimes q})$ where $\mathcal{P}$ is O(3)-equivariant projection. Hence

$$\mathbf{B}(R\mathbf{n}) = \mathcal{P}\big((R\mathbf{n})^{\otimes q}\big) = \mathcal{P}\big(R^{\otimes q}\mathbf{n}^{\otimes q}\big) = R^{\otimes q}\mathcal{P}\big(\mathbf{n}^{\otimes q}\big) = \rho_q(R)\mathbf{B}(\mathbf{n}). \tag{21}$$
$\square$

**Lemma A.2** (Equivariance of the environment tensors). *Let $\mathbf{E}_{i,k}$ be defined as a neighbor sum of $\mathbf{B}(\mathbf{n}_{ij})$ weighted by scalar coefficients $\alpha_{ij,k}$ produced by the invariant backbone. Then for all $R \in \mathrm{O}(3)$,*

$$\mathbf{E}_{i,k}(R \cdot G) \;=\; \rho_q(R)\, \mathbf{E}_{i,k}(G). \tag{22}$$

*Proof.* Since $\alpha_{ij,k}$ are scalars from an invariant backbone, $\alpha_{ij,k}(R \cdot G) = \alpha_{ij,k}(G)$. Using Lemma A.1,

$$\mathbf{E}_{i,k}(R \cdot G) = \sum_j \alpha_{ij,k}(G)\mathbf{B}(R\mathbf{n}_{ij}) = \sum_j \alpha_{ij,k}(G)\rho_q(R)\mathbf{B}(\mathbf{n}_{ij}) = \rho_q(R)\mathbf{E}_{i,k}(G).$$

$\square$

**Lemma A.3** (Equivariance of the tensor head). *Let $\mathcal{T}_i$ denote the atom-wise tensor predicted by the head from $\{\mathbf{E}_{i,k}\}_k$ (as defined in the main text). Then for all $R \in \mathrm{O}(3)$,*

$$\mathcal{T}_i(R \cdot G) \;=\; \rho_r(R)\,\mathcal{T}_i(G). \tag{23}$$

*Proof.* The head is a composition of channel-only linear mixing with scalar parameters and tensor operations that are $\mathrm{O}(3)$-equivariant, including outer products and dot product. For channel-only linear mixing, if $\tilde{E}_{i,k} := \sum_\ell M_{k\ell} E_{i,\ell}$, then by Lemma A.2,

$$\tilde{E}_{i,k}(R \cdot G) = \sum_\ell M_{k\ell} E_{i,\ell}(R \cdot G) = \sum_\ell M_{k\ell} \rho_q(R) E_{i,\ell}(G) = \rho_q(R)\tilde{E}_{i,k}(G). \tag{24}$$

The rest of the head only applies dot/outer products, plus linear combinations with scalar coefficients. Hence the remaining map is $\mathrm{O}(3)$-equivariant. Therefore $\mathcal{T}_i(R \cdot G) = \rho_r(R)\mathcal{T}_i(G)$. $\qquad\square$

**Lemma A.4** (Equivariance of pooling). *Let the crystal-level prediction be obtained by pooling, $\mathcal{T} := \sum_i \alpha_i \mathcal{T}_i$. Then for all $R \in \mathrm{O}(3)$,*

$$\mathcal{T}(R \cdot G) \;=\; \rho_r(R)\,\mathcal{T}(G). \tag{25}$$

*Proof.* By linearity of pooling and Lemma A.3,

$$\mathcal{T}(R \cdot G) = \sum_i \alpha_i \mathcal{T}_i(R \cdot G) = \sum_i \alpha_i \rho_r(R)\mathcal{T}_i(G) = \rho_r(R) \sum_i \alpha_i \mathcal{T}_i(G) = \rho_r(R)\mathcal{T}(G). \tag{26}$$
$$\square$$

**Theorem A.5** ($\mathrm{O}(3)$-equivariance of CEITNet). *For any $r \in \{2, 3, 4\}$ and any $R \in \mathrm{O}(3)$, CEITNet satisfies (19).*

*Proof.* By Lemma A.2, each environment tensor transforms according to its own order, i.e., by $\rho_q(R)$. By Lemmas A.3 and A.4, the task-specific head maps these equivariant environment tensors to an order-$r$ tensor, and pooling preserves this transformation, yielding (19). $\qquad\square$

### A.8. Computational Complexity

Spherical-harmonic equivariant methods propagate features in irreducible representations and combine them through Clebsch–Gordan tensor products. When irreducible representations are retained up to maximum degree $L$, the full tensor-product coupling can incur high-order cost, commonly on the order of $\mathcal{O}(L^6)$ in the unrestricted case.

In contrast, CEITNet keeps learnable transformations in channel space and uses Cartesian equivariant bases only when assembling the tensor head. For a graph with $N$ atoms, average degree $d$, channel width $K$, and basis set size $|\mathcal{B}|$, the local-environment construction costs $\mathcal{O}(NdK|\mathcal{B}|)$, and the two-stream channel coupling costs $\mathcal{O}(NK^2)$ per tensor head. This decoupled design avoids propagating high-order equivariant features through the entire backbone and avoids expensive Clebsch–Gordan tensor products.

