# OpenReview forum: "Efficient Equivariant High-Order Crystal Tensor Prediction via Cartesian Local-Environment Many-Body Coupling"
_ICML.cc/2026/Conference — ICML 2026 regular_

### Official Review · Reviewer_4Uhf · 2026-03-09

**Soundness:** 3
**Presentation:** 3
**Significance:** 2
**Originality:** 2
**Overall Recommendation:** 4
**Confidence:** 3

**Summary:**

This paper proposes CEITNet, an efficient equivariant architecture for predicting high-order crystal tensor properties using Cartesian local environment coupling. The method avoids expensive spherical-harmonic tensor products and shows improved efficiency and accuracy on dielectric, piezoelectric, and elastic tensor benchmarks. The approach is technically sound and the results are competitive, but several issues regarding generality, physical consistency, and evaluation realism remain unclear.

**Compliance With Llm Reviewing Policy:**

Affirmed.

**Final Justification:**

I checked all my concerns, but still think the same about total scores. Thank you for your good work.

**Key Questions For Authors:**

1. Does the symmetry correction step use ground-truth space-group information? How robust is the method when the structure is distorted or the symmetry is ambiguous?
2. Are the results in Tables 4–6 reported before or after symmetry correction?
3. How difficult is it to extend CEITNet to new tensor types beyond the three studied here?
4. Can the authors provide experiments on more challenging splits, such as composition split or symmetry split?
5. For the elastic benchmark, how were the reimplemented baselines tuned and verified?

**Limitations:**

yes

**Strengths And Weaknesses:**

# Strengths

- The paper addresses a critical bottleneck in material science by predicting high-order tensors that are computationally expensive for DFT.
- The proposed architecture achieves a significant speedup and reduces parameter counts compared to baselines.
- The method achieves competitive results on multiple tensor prediction tasks, consistently outperforming GMTNet and GeoCTP in precision metrics.
- The paper is generally well written and experiments are reasonably thorough.

---

# Weaknesses

## Major Weakness 1 — Limited generality of the proposed framework

The method is not fully task-agnostic. The authors acknowledge this limitation, but it still weakens the generality claim. Each tensor type requires a manually designed coupling rule (e.g., dot product, outer product, vector–matrix coupling). The paper itself acknowledges additional design effort when transferring to new tensor types. This weakens the claim that CEITNet is a general framework for arbitrary high-order tensor prediction.

## Major Weakness 2 — Crystal symmetry is not fully enforced inside the model

O(3) equivariance is built into the architecture, but crystal space-group symmetry is handled by a post-hoc correction step. Table 9 shows that symmetry constraints are not always satisfied before correction. It is unclear whether the main benchmark tables use corrected or uncorrected outputs. The reliance on post-processing weakens the claim of intrinsic physical consistency.

## Major Weakness 3 — Limited evaluation of generalization

All experiments use standard IID splits on relatively small datasets. No tests are provided for out-of-distribution structues. Given that tensor properties strongly depend on crystal symmetry and composition, it would be important to test generalization across crystal systems, compositions, or structure families. Without such experiments, it is unclear whether the model learns transferable physical structure or only fits the benchmark distribution.

---

## Minor Weakness 1 — Piezoelectric benchmark removes zero-tensor samples

Zero-tensor samples are removed to avoid trivial predictors. This is reasonable for metric stability, but it also removes the practical screening task of identifying whether a material has non-zero response. The current evaluation focuses only on conditional regression.

## Minor Weakness 2 — Baseline comparison relies on reimplementation

For the elastic task, some baselines are re-evaluated by the authors. This may be necessary, but the paper provides limited detail on tuning and verification. Additional sanity checks would improve confidence in the comparison.

## Minor Weakness 3 — Efficiency comparison is limited

Runtime comparison is reported only against GMTNet, while other baselines are not included. A broader comparison would strengthen the efficiency claim.

---

> ### Author Rebuttal · Authors · 2026-03-31
>
> Thank you for the constructive feedback. We respond point-by-point below, and will incorporate the discussions in the revised manuscript.
>
> **Q1.** We do not use ground-truth. Instead, we follow the GMTNet method to infer symmetry constraints from the inputs. This step is used to enforce exact symmetry-induced constraints in the predicted tensor (see our response to Q2/Major W2 below). For perfect crystals, this correction is useful for improving exact physical consistency. For non-ideal or distorted structures, the raw network prediction can be reported directly (the correction should be viewed as an optional refinement rather than a requirement).
>
> **Q2/Major W2.** Tables 4–6 report symmetry-corrected outputs for fair comparison with prior work. We agree that exact crystal space-group constraints are not fully enforced inside the network, but Table 9 shows that without correction, our proposed CEITNet already satisfies symmetry-enforced zeros better than previous methods. In CEITNet, equivariance and the intrinsic symmetries of the target tensor are guaranteed by construction. These additional constraints are used as a final post-processing refinement step. Without this step, the raw predictions may contain small numerical residuals on symmetry-forbidden components, or slight mismatches between components that should be equal.
>
> **Q3/Major W1.** Thanks for your suggestions. Extending CEITNet to a new tensor type does require additional design, but this cost is localized to the output/coupling head rather than requiring a redesign of the full model. As the experiments show, this framework already works well across dielectric, piezoelectric, and elastic prediction, covering order-2, order-3, and order-4 tensor targets. A more automated way of constructing such target-specific tensor heads would be an interesting direction for future work. We will add some discussion in the revised manuscript.
>
> **Q4/Major W3.** We conducted OOD tests (leave-one-system-out) on the dielectric benchmark. Results below show that CEITNet consistently outperforms GMTNet across all six held-out systems. This additional OOD test on the dielectric benchmark suggests better cross-system transfer for CEITNet.
>
> |CEITNet|Fnorm↓|EwT 25%↑|EwT 10%↑|EwT 5%↑|
> |-|-:|-:|-:|-:|
> |Cubic|10.7|64.6%|37.9%|20.7%|
> |Orthorhombic|3.6|78.2%|41.8%|17.2%|
> |Hexagonal|5.7|80.7%|56.9%|29.8%|
> |Trigonal|5.6|81.1%|56.2%|29.5%|
> |Monoclinic|3.3|83.6%|49.8%|21.6%|
> |Tetragonal|4.1|85.0%|47.9%|20.5%|
>
> |GMTNet|Fnorm↓|EwT 25%↑|EwT 10%↑|EwT 5%↑|
> |-|-:|-:|-:|-:|
> |Cubic|11.1|61.5%|31.5%|16.1%|
> |Orthorhombic|4.2|71.9%|38.9%|13.0%|
> |Hexagonal|6.6|76.8%|49.1%|22.3%|
> |Trigonal|5.8|77.7%|33.0%|13.2%|
> |Monoclinic|3.4|79.3%|43.4%|14.7%|
> |Tetragonal|4.3|82.4%|42.8%|17.6%|
>
> **Q5/Minor W2.** For our own implementation and runtime benchmarking, we followed Hua et al. using the Yan et al. codebase and kept the same split, graph construction, backbone setting, training budget, and evaluation pipeline. We will add these implementation details more explicitly in the revision.
>
> **Minor W1.** We agree that removing zero-tensor samples changes the task from full screening to conditional regression. We adopt this setting because, on the original dataset, the benchmark becomes substantially degenerate: as shown in Appendix A.2 Table 10, a trivial all-zero predictor already achieves strong EwT scores (45.7/45.7/45.7 for EwT 25%/EwT 10%/EwT 5%, compared to GMTNet (with inferred zeros) 49.1/46.3/45.7), indicating that the metric is heavily dominated by zero samples. We therefore follow the GeoCTP preprocessing protocol to remove zero samples for a more informative regression benchmark. For completeness, we also evaluated on the original unfiltered dataset (both methods without inferred zeros): our method achieves 37.1/29.1/27.3, while GMTNet achieves 27.5/26.1/25.9.
>
>
> **Minor W3.** We have now additionally benchmarked the runtime of GeoCTP. For Dielectric/Piezoelectric/Elastic, the results are shown below. Overall, CEITNet remains faster than GMTNet and shows comparable inference efficiency to GeoCTP across the three benchmarks.
>
> |Runtime(s)↓|Diele|Piezo|Elast|
> |-|-:|-:|-:|
> |GMTNet|21|50|616|
> |GeoCTP|6|5|50|
> |CEITNet|5|6|46|
>
> |Throughput(item/s)↑|Diele|Piezo|Elast|
> |-|-:|-:|-:|
> |GMTNet|22.02|5.40|2.31|
> |GeoCTP|69.61|52.08|28.39|
> |CEITNet|85.90|40.64|30.79|

---

> > ### Author Rebuttal · Reviewer_4Uhf · 2026-04-01
> >
> > Thank you. This response resolved all my concerns.

---

> > > ### Author Response · Authors · 2026-04-02
> > >
> > > Dear Reviewer 4Uhf,
> > >
> > > Thank you for the constructive feedback that helped improve our paper. As you kindly confirmed that your concerns have been resolved, we hope that our clarifications and additional results will be helpful for your final assessment.
> > >
> > > Thanks again for your time and continued engagement with our paper.

---

### Official Review · Reviewer_hHjF · 2026-03-11

**Soundness:** 3
**Presentation:** 3
**Significance:** 3
**Originality:** 2
**Overall Recommendation:** 4
**Confidence:** 3

**Summary:**

This paper targets the excessive computational and memory costs incurred by Clebsch-Gordan tensor products in spherical-harmonic-based equivariant models for end-to-end prediction of high-order tensor properties in crystals. To address this issue, it proposes the Cartesian Environment Interaction Tensor Network. The model constructs a multi-channel Cartesian local-environment tensor for each atom and performs learnable many-body coupling in the channel space. By combining invariant message passing with Cartesian tensor bases, it enables the efficient construction of equivariant high-order tensors.

**Compliance With Llm Reviewing Policy:**

Affirmed.

**Key Questions For Authors:**

The Cartesian basis set is manually chosen. How can the authors ensure that this design generalizes to other high-order tensor properties or new crystal systems?
Nearly 50% of zero-tensor samples are removed from the piezoelectric dataset, which may bias the evaluation. How do the authors justify this preprocessing choice, and what is the model performance on the original unfiltered dataset?

**Limitations:**

1) The weighted aggregation formulation of the channelized local environment tensors lacks theoretical justification, and it is not explained why it can preserve the expressive power for high-order tensors.
2) The Cartesian basis set is manually selected, and neither the completeness of the basis nor its applicability to diverse crystal systems has been proven.
3) Experiments are only conducted on the JARVIS dataset, and the generalization ability has not been verified on external datasets such as Materials Project and OQMD.
4) The paper attributes the computational bottleneck of spherical-harmonic equivariant models to Clebsch-Gordan tensor products, yet it does not provide a quantitative complexity comparison between Clebsch-Gordan tensor products and the proposed Cartesian tensor operations across different tensor orders. Furthermore, it fails to explain why the proposed Cartesian tensor operations can both preserve the tensor spatial structure of crystalline materials and achieve improved computational speed.

**Strengths And Weaknesses:**

Strengths:
It proposes a paradigm that decouples invariant encoding from equivariant tensor construction. It achieves faster inference speed than GMTNet with fewer parameters, delivering superior performance in parameter efficiency and computational efficiency.

Weaknesses:
1) The weighted aggregation formulation of the channelized local environment tensors lacks theoretical justification, and it is not explained why it can preserve the expressive power for high-order tensors.
2) The Cartesian basis set is manually selected, and neither the completeness of the basis nor its applicability to diverse crystal systems has been proven.
3) Experiments are only conducted on the JARVIS dataset, and the generalization ability has not been verified on external datasets such as Materials Project and OQMD.
4) The paper attributes the computational bottleneck of spherical-harmonic equivariant models to Clebsch-Gordan tensor products, yet it does not provide a quantitative complexity comparison between Clebsch-Gordan tensor products and the proposed Cartesian tensor operations across different tensor orders. Furthermore, it fails to explain why the proposed Cartesian tensor operations can both preserve the tensor spatial structure of crystalline materials and achieve improved computational speed.

---

> ### Author Rebuttal · Authors · 2026-03-31
>
> Thank you for the constructive feedback. We respond point-by-point below, and will incorporate the discussions in the revised manuscript.
>
> **W1.** The weighted aggregation in Eq. (6) is the step that lifts invariant atomic context into a multi-channel equivariant local-environment representation. Concretely, the weights $w_{ij,k}$ are generated by an MLP from the invariant context $z_{ij}$, while $B_{ij}$ is constructed from Cartesian equivariant bases, thus $E_{i,k}$ is permutation-invariant and equivariant by construction. We have provided a formal equivariance proof in Appendix A.4.
>
> In terms of expressivity, Eq. (6) generates channelized local environments, combined by the interaction head to achieve the coupling of two pooled environments, which naturally introduces cross-neighbor terms corresponding to effective three-body interactions without explicitly enumerating triplets. Thus, this design offers an expressive and efficient structure for the tensor tasks considered in this work.
>
> **W2/Q1.** We will clarify the scope more carefully. For transfer across crystal systems, our Cartesian basis is not designed for any specific crystal family. Instead, it is built from general geometric building blocks of the local environment. To further test whether this design transfers across crystal systems, we additionally ran leave-one-system-out experiments on the dielectric benchmark below. Our proposed CEITNet consistently outperforms GMTNet on all six held-out crystal systems, which supports that the basis design is not tied to the crystal systems seen during training.
>
> For transfer to other high-order tensor properties, our view is that the same geometric primitives provide a constructive extension basis: higher-order Cartesian tensor products and a target-specific head can be instantiated according to the order and intrinsic symmetry of the target tensor. At the same time, we agree that a more automated and systematic way of construction would be an interesting direction for future work, and we will discuss this in the revised manuscript.
>
> |CEITNet|Fnorm↓|EwT 25%↑|EwT 10%↑|EwT 5%↑|
> |-|-:|-:|-:|-:|
> |Cubic|10.7|64.6%|37.9%|20.7%|
> |Orthorhombic|3.6|78.2%|41.8%|17.2%|
> |Hexagonal|5.7|80.7%|56.9%|29.8%|
> |Trigonal|5.6|81.1%|56.2%|29.5%|
> |Monoclinic|3.3|83.6%|49.8%|21.6%|
> |Tetragonal|4.1|85.0%|47.9%|20.5%|
>
> |GMTNet|Fnorm↓|EwT 25%↑|EwT 10%↑|EwT 5%↑|
> |-|-:|-:|-:|-:|
> |Cubic|11.1|61.5%|31.5%|16.1%|
> |Orthorhombic|4.2|71.9%|38.9%|13.0%|
> |Hexagonal|6.6|76.8%|49.1%|22.3%|
> |Trigonal|5.8|77.7%|33.0%|13.2%|
> |Monoclinic|3.4|79.3%|43.4%|14.7%|
> |Tetragonal|4.3|82.4%|42.8%|17.6%|
>
> **W3.** We add experiments on the Materials Project dielectric benchmark. Both models were evaluated under the same backbone and metric pipeline. These results below show that CEITNet also outperforms GMTNet on an external benchmark. We will include the dataset details and protocol in the revised manuscript.
>
> |Model|Fnorm↓|EwT 25%↑|EwT 10%↑|EwT 5%↑|
> |-|-:|-:|-:|-:|
> |CEITNet|0.1884|100.00%|91.40%|62.37%|
> |GMTNet|0.2023|100.00%|89.25%|58.06%|
>
>
> **W4.** For a fair comparison, we focus on the non-shared tensor-construction and output part, since CEITNet and GMTNet use the same invariant backbone. GMTNet relies on spherical-harmonic irreps and Clebsch–Gordan couplings to form tensor-valued features, so its extra cost  is not only tensor-order dependent but also path dependent. Let $n=|V|$, $e=|E|$, $K$ be the number of environment channels, and $d_r$ denotes the effective Cartesian tensor dimension associated with an order-$r$ target. In CEITNet, the channelized local-environment construction costs $O(eKd_r)$ and the channel-space interaction costs $O(nK^2d_r)$, giving coarse complexity bound $O(eKd_r+nK^2d_r).$ CEITNet directly constructs the target tensor in the Cartesian basis and explicitly enforces the required physical symmetries. Its efficiency gain therefore does not come from simplifying the tensor structure, but from avoiding CG-based tensor couplings while still preserving the spatial structure of the material tensor. The result is consistent with Table 7, where the inference times of GMTNet/CEITNet are 21/5 s, 50/6 s, and 616/46 s for dielectric, piezoelectric, and elastic prediction, respectively.
>
> **Q2.** We follow the zero-excluded setting used in Hua et al.(2025). One key reason is, on the original dataset, the benchmark becomes substantially degenerate. As shown in Appendix A.2 Table 10, a trivial all-zero predictor already achieves strong EwT scores (45.7/45.7/45.7 for EwT 25%/EwT 10%/EwT 5%, compared to GMTNet (with inferred zeros) 49.1/46.3/45.7), indicating that the metric is heavily dominated by zero samples. We therefore follow the GeoCTP preprocessing protocol for a more informative regression benchmark. For completeness, we also evaluated on the original unfiltered dataset (both methods without inferred zeros): our method achieves 37.1/29.1/27.3, while GMTNet achieves 27.5/26.1/25.9.

---

> > ### Author Rebuttal · Reviewer_hHjF · 2026-04-01
> >
> > The test dataset size is still small.

---

> > > ### Author Response · Authors · 2026-04-02
> > >
> > > Dear Reviewer hHjF,
> > >
> > > Thank you for your follow-up comment.
> > >
> > > The limited availability of large-scale benchmarks for high-order tensor properties is a common challenge in this area, mainly because obtaining such labels is expensive.
> > > Under this practical constraint, our main evaluation is conducted on the three benchmark datasets (dielectric, piezoelectric, and elastic) used in prior work (e.g., Yan et al., Hua et al.) to ensure direct and fair comparisons.
> > >
> > > In addition, we further included experiments on the Materials Project dielectric benchmark and leave-one-system-out experiments across six crystal systems, where CEITNet consistently shows strong performance. These additional results suggest that the observed gains are not tied to one particular split or dataset.
> > >
> > > We will clarify this limitation more explicitly in the limitations section of the revised manuscript. Thank you again for your continued engagement with our work.

---

### Official Review · Reviewer_jq7Q · 2026-03-12

**Soundness:** 3
**Presentation:** 3
**Significance:** 3
**Originality:** 3
**Overall Recommendation:** 4
**Confidence:** 4

**Summary:**

CEITNet predicts order-2, 3, and 4 crystal tensors by replacing CG tensor products with a Cartesian local-environment construction: neighbor directions are projected onto geometric bases (unit vectors, the traceless deviatoric Q, identity), aggregated into K channels per atom, then coupled via a learnable K×K interaction matrix to produce the output tensor. An invariant ComformerConv backbone handles chemistry; the equivariant head is entirely separate. The method is evaluated on JARVIS-DFT dielectric, piezoelectric, and elastic benchmarks, showing accuracy gains over GMTNet and GeoCTP alongside large inference speedups.

**Compliance With Llm Reviewing Policy:**

Affirmed.

**Key Questions For Authors:**

1.Table 6 shows CEITNet's Fnorm is worse than GMTNet on the elastic task (70.11 vs. 67.38). Is this a distributional issue, i.e., does CEITNet improve on most samples but fail badly on a subset of hard outliers? An error percentile breakdown would clarify whether this is a meaningful reliability gap or a few catastrophic predictions. The answer would affect how I interpret the EwT gains.

2.GeoCTP's EwT 5% on piezoelectric is 0.04%, far below GMTNet's 1.11%, despite sharing the same backbone and preprocessing. Were GeoCTP's reported numbers produced under the same zero-excluded split and training budget as your model, or carried over from a different setting in Hua et al. (2025)? This matters because CEITNet's largest margins are against an apparently undertuned baseline.

3.Dong et al. (2025) is cited in Section 4.2 as a competing end-to-end equivariant piezoelectric model but is absent from Table 5. Why? If the dataset differs, say so. If it simply wasn't evaluated, that's the most relevant missing comparison in the paper.

**Limitations:**

The conclusion identifies task-specific coupling design and the three-body interaction ceiling as limitations. What's missing: no acknowledgment that the Fnorm regression on elastic relative to GMTNet is a potential concern for applications sensitive to worst-case errors.

**Strengths And Weaknesses:**

Table 7 shows a 4x speedup on dielectric and a 13x speedup on elastic inference versus GMTNet, with parameter count dropping from 1.1M (ETGNN) to 0.6M. Those are real, well-documented gains, not marginal ones. The piezoelectric jump in Table 5 is striking: EwT 25% goes from GeoCTP's 2.59% to CEITNet's 21.98%. But GeoCTP's piezoelectric numbers look anomalously weak. Its EwT 5% is 0.04%, worse than GMTNet's 1.11% despite sharing the same backbone. The authors note these come from Hua et al. (2025), but give no explanation. If GeoCTP wasn't properly tuned on the zero-excluded split, the margin is partially an artifact, not a fair reflection of method quality.

On elastic (Table 6), CEITNet's Fnorm is worse than GMTNet. The authors call this competitive, which is technically defensible given the EwT improvements, but Fnorm and EwT capture different failure modes. Fnorm is dominated by hard outliers. The paper glosses over this rather than addressing it directly. The ablation in Table 8 is dielectric-only for the K sweep and the sep/inter decomposition. Table 12 adds w/o mix for piezoelectric and elastic, but that's it. Whether K=16 and the coupling design choices generalize to order-3 and order-4 tasks is simply not shown. Dong et al. (2025) is cited in Section 4.2 as an end-to-end equivariant piezoelectric method but never appears in Table 5. That's the one missing baseline that would most sharpen the piezoelectric comparison.

---

> ### Author Rebuttal · Authors · 2026-03-31
>
> Thank you for the constructive feedback. We respond point-by-point below, and will incorporate the discussions in the revised manuscript.
>
> **Q1/W2.** We agree that Fnorm and EwT capture different failure modes. Since the GMTNet checkpoint is not publicly available, we are unable to perform a direct per-sample comparison between the two models. Here, we conducted a per-sample analysis of our CEITNet on the elastic test set: the median sample-wise Frobenius error is 49.0, while the 95th percentile is 233.5 and the worst 5% of samples contribute 28.8% of the total error. These results indicate that CEITNet’s mean Fnorm is substantially influenced by 5% tail subset.
>
> **Q2/W1.** The piezoelectric GeoCTP results in Table 5 were not produced by our reimplementation. Instead, we directly report the results of the GeoCTP and GMTNet in Hua et al. (2025). According to Hua et al., these results were obtained under the zero-excluded piezoelectric protocol and using the same benchmark codebase and setting as ours.
>
> **Q3/W4.** Dong et al. (2025) is conceptually relevant as an end-to-end equivariant piezoelectric predictor. We did not include it in Table 5 because Dong et al. (2025) was evaluated on a different dataset and under a different evaluation protocol, and is therefore not numerically comparable to our setting.
>
> **W3.** We additionally performed extra K sweeps on both piezoelectric and elastic. These results below show that K=16 remains slightly better overall on elastic. On piezoelectric, K=32 is slightly better on some metrics. We therefore view K = 16 as a default rather than an optimal setting for every task. In addition, the w/o inter/sep ablations show a consistent performance drop across both tasks, indicating that both components are important for our model across tasks.
>
> |Piezo|Fnorm↓|EwT 25%↑|EwT 10%↑|EwT 5%↑|
> |-|-:|-:|-:|-:|
> |K=16|0.517|21.98%|5.80%|2.72%|
> |K=8|0.536|21.48%|4.07%|1.11%|
> |K=32|0.514|20.00%|5.19%|1.85%|
> |w/o inter|0.567|14.07%|4.07%|1.11%|
> |w/o sep|0.693|6.67%|1.85%|1.48%|
>
> |Elast|Fnorm↓|EwT 25%↑|EwT 10%↑|EwT 5%↑|
> |-|-:|-:|-:|-:|
> |K=16|70.11|70.6%|32.2%|14.4%|
> |K=8|71.23|69.90%|31.50%|14.56%|
> |K=32|72.73|69.34%|29.89%|12.80%|
> |w/o inter|72.23|69.34%|29.61%|12.03%|
> |w/o sep|85.65|57.74%|17.65%|7.03%|

---

> > ### Author Rebuttal · Reviewer_jq7Q · 2026-04-04
> >
> > I have read the rebuttal. My original review posed questions to the authors, which the rebuttal has now addressed. I have posted my detailed post-rebuttal assessment as an Official Comment.

---

> > > ### Author Response · Authors · 2026-04-04
> > >
> > > Dear Reviewer jq7Q,
> > >
> > > Thank you for your continued engagement with our paper.
> > >
> > > If you have any remaining concerns, we would greatly appreciate it if you could share them in the Rebuttal Acknowledgemen, as the Official Comment is not visible to us. We would be glad to address them and further improve the paper.
> > >
> > > Thank you again for your time and consideration.

---

### Official Review · Reviewer_ZJJ1 · 2026-03-18

**Soundness:** 3
**Presentation:** 3
**Significance:** 3
**Originality:** 3
**Overall Recommendation:** 4
**Confidence:** 4

**Summary:**

The paper proposes CEITNet to predict high-order crystal tensor properties. Instead of using spherical harmonics and Clebsch-Gordan tensor product, the authors use Cartesian geometric bases to form channelized local environment tensors and build the equivariant interaction via channel mixing. Results show the effectiveness on the JARVIS-DFT dataset.

**Compliance With Llm Reviewing Policy:**

Affirmed.

**Final Justification:**

The rebuttal has addressed my concerns, and the methodology is clear and informative, although the technique is similar to the irreducible Cartesian tensor product.

**Key Questions For Authors:**

See weaknesses.

**Limitations:**

Yes.

**Strengths And Weaknesses:**

Strengths:

- The paper is clearly written and well demonstrated, and the local environment tensor idea appears novel to me.
- The results are consistently better than previous baselines, including GMTNet.

Weaknesses:

- The dataset size is small. Since it follows the same setting as GMTNet, I do not think this diminishes the value of the paper.
- The methodology is very close to low-order irreducible Cartesian tensors (ICTs). For example, under your notation, $n$ corresponds to the order-1 ICT and $Q$ corresponds to the order-2 ICT. It would be helpful to discuss and cite [1] [2]. Also, since only channel mixing is used rather than irreps mixing, it is recommended to show whether incorporating tensor products of ICTs could further improve performance.
- The title suggests many-body but the formulation is three-body. I recommend a general formula with many-body then restrict to three-body for the writing.

[1] Higher-Rank Irreducible Cartesian Tensors for Equivariant Message Passing. Zaverkin., et al.

[2] High-Rank Irreducible Cartesian Tensor Decomposition and Bases of Equivariant Spaces. Shao., et al.

---

> ### Author Rebuttal · Authors · 2026-03-31
>
> Thank you for your constructive suggestions. We respond point-by-point below, and will incorporate the discussions in the revised manuscript.
>
> **W1.** Thanks for your comments. We agree that the current benchmark size is relatively small. One reason is that reliable labels for tensor prediction are expensive to obtain.
> We will clarify this scope more explicitly in the revision.
>
> **W2.** Thanks for pointing out the relevance of [1] and [2]. We agree that our Cartesian bases are related to the ICT formalism, and we will state it explicitly in the revised manuscript by citing and discussing these works. Here, we briefly compare our work to the two references you mentioned.
>
> Paper [1] introduces ICTs into MPNNs and improves computational efficiency. However, its primary focus is on the prediction of energies and atomic forces rather than high-order tensors. Instead, one of the key contributions of our paper is introducing the proposal of a channelized local-environment construction together with a corresponding prediction head, where the learnable transformations are kept in channel space, thereby maintaining flexibility while remaining efficient. For the second reference paper [2], its decomposition of Cartesian tensors provides a valuable tool for our future extensions.
>
> Regarding the reviewer’s suggestion to explore whether incorporating tensor products of ICTs could further improve performance, we agree that this is a promising idea to further enhance the expressive power of the model. One concern is it may introduce additional computational overhead, which is one of the key trade-offs to balance in our design. In the revised manuscript, we will discuss this idea as a potential future research direction.
>
> **W3.** Thanks for your suggestions. We agree that it will be better to introduce the general formula first. In the revised manuscript, we will follow your suggestion to first present a more general formulation, $T_i=A\left(E_i^{\tau_0}\right)+\sum_{q=2}^Q\gamma_q\psi_q\left(E_i^{\tau_{1}}, \ldots, E_i^{\tau_{q}}; M^{(q)}  \right)+\Delta T_i.$ We will then restrict it to the three-body setting (i.e., taking Q=2).

---

> > ### Author Rebuttal · Reviewer_ZJJ1 · 2026-04-02
> >
> > Thank you for your rebuttal and my concerns have all been addressed.

---

> > > ### Author Response · Authors · 2026-04-03
> > >
> > > Dear Reviewer ZJJ1,
> > >
> > > Thank you for the constructive feedback and suggestions, which have helped improve our paper. We are glad that our response resolved your concerns, and we would be very grateful if you could update your review to reflect your current assessment.
> > >
> > > Thank you for your time and continued engagement.

---

### Decision · Program_Chairs · 2026-04-30

**Decision:**

Accept (regular)

**Comment:**

This paper proposes CEITNet, which predicts high-order crystal tensor properties (order-2 dielectric, order-3 piezoelectric, order-4 elastic) by constructing multi-channel Cartesian local environment tensors and coupling them via a learnable channel-space interaction matrix. This avoids the expensive Clebsch-Gordan tensor products of spherical-harmonic approaches while maintaining O(3) equivariance.

Reviewers converged on acceptance. Concerns center on limited generality (task-specific coupling heads), small benchmark scale, incomplete baselines, and the Fnorm regression on elastic relative to GMTNet. The rebuttal provided additional experiments (Materials Project benchmark, leave-one-system-out, K sweeps on all tasks, GeoCTP runtime) that addressed the concerns.